# Flecainide Paradoxically Activates Cardiac Ryanodine Receptor Channels under Low Activity Conditions: A Potential Pro-Arrhythmic Action

**DOI:** 10.3390/cells10082101

**Published:** 2021-08-16

**Authors:** Samantha C. Salvage, Esther M. Gallant, James A. Fraser, Christopher L.-H. Huang, Angela F. Dulhunty

**Affiliations:** 1Physiological Laboratory, University of Cambridge, Downing Street, Cambridge CB2 3EG, UK; ss2148@cam.ac.uk (S.C.S.); jaf21@cam.ac.uk (J.A.F.); clh11@cam.ac.uk (C.L.-H.H.); 2Department of Biochemistry, University of Cambridge, Tennis Court Road, Cambridge CB2 1QW, UK; 3Eccles Institute of Neuroscience, John Curtin School of Medical Research, The Australian National University, 131 Garran Road, Acton 2601, Australia; galla001@umn.edu

**Keywords:** RyR2, RyR2-P2328S, flecainide, catecholaminergic polymorphic ventricular tachycardia, atrial fibrillation, RyR2 activation, RyR2 inhibition

## Abstract

Cardiac ryanodine receptor (RyR2) mutations are implicated in the potentially fatal catecholaminergic polymorphic ventricular tachycardia (CPVT) and in atrial fibrillation. CPVT has been successfully treated with flecainide monotherapy, with occasional notable exceptions. Reported actions of flecainide on cardiac sodium currents from mice carrying the pro-arrhythmic homozygotic RyR2-P2328S mutation prompted our explorations of the effects of flecainide on their RyR2 channels. Lipid bilayer electrophysiology techniques demonstrated a novel, paradoxical increase in RyR2 activity. Preceding flecainide exposure, channels were mildly activated by 1 mM luminal Ca^2+^ and 1 µM cytoplasmic Ca^2+^, with open probabilities (*P_o_*) of 0.03 ± 0.01 (wild type, WT) or 0.096 ± 0.024 (P2328S). Open probability (*P_o_*) increased within 0.5 to 3 min of exposure to 0.5 to 5.0 µM cytoplasmic flecainide, then declined with higher concentrations of flecainide. There were no such increases in a subset of high *P_o_* channels with *P_o_* ≥ 0.08, although *P_o_* then declined with ≥5 µM (WT) or ≥50 µM flecainide (P2328S). On average, channels with *P_o_* < 0.08 were significantly activated by 0.5 to 10 µM of flecainide (WT) or 0.5 to 50 µM of flecainide (P2328S). These results suggest that flecainide can bind to separate activation and inhibition sites on RyR2, with activation dominating in lower activity channels and inhibition dominating in more active channels.

## 1. Introduction

Normal cardiac function depends upon regulatory surface membrane ion channel and transporter activity and the resulting Ca^2+^ release and reuptake processes between the cytosol and the intracellular sarcoplasmic reticular (SR) Ca^2+^ store in its participating cardiac myocytes. Mutations in proteins underlying these processes can lead to arrhythmic conditions and cardiac failure [1]. Of these, mutations in the cardiac sarcoplasmic reticulum (SR) ryanodine receptor (RyR2) Ca^2+^ release channel can increase channel opening during diastole, with a consequent excess SR Ca^2+^ release. Cardiac ryanodine receptor (RyR2) mutations have been associated with catecholaminergic polymorphic ventricular tachycardia (CPVT) [2], ventricular fibrillation (VF) [3] and/or atrial fibrillation (AF) [4]. Recent studies reported that RyR2 channels from RyR2-P2328S mouse hearts modelling a clinical CPVT variant showed increased activity at diastolic Ca^2+^ concentrations compared to wild-type (WT) RyR2 channels [5]. This increased activity parallels pro-arrhythmic effects of the mutation in mouse models and humans and has similar features to other arrhythmogenic RyR2 mutations [6,7,8,9,10].

The increases in diastolic cytosolic Ca^2+^ concentrations associated with these mutations may activate surface membrane sodium calcium exchanger (NCX) activity. Its electrogenic effects from extruding 1 Ca^2+^ in exchange for 3 Na^+^ ions can generate delayed after-depolarisation (DAD) phenomena that can trigger pro-arrhythmic ectopic action potentials, if reaching threshold voltage. Elevated cytosolic Ca^2+^ concentrations have also been implicated in reducing sodium channel (Na_V_) activity [11,12]. This would compromise action potential generation and conduction, producing a pro-arrhythmic re-entrant substrate. Mouse RyR2-P2328S hearts demonstrated reduced Na_v_1.5 currents (*I*_Na_) and slowed action potential conduction velocities [7,13,14,15]. These pro-arrhythmic properties were reduced by the established, anti-arrhythmic drug flecainide [7].

Flecainide is an established class Ic anti-arrhythmic Nav1.5 blocking agent. Yet, it paradoxically rescued the reductions in *I*_Na_ and action potential wavelength in RyR2-P2328S hearts, findings unexpected for a Na_v_1.5 blocker. Therefore, these findings may reflect actions on other components of the excitation-contraction coupling pathway, such as its more recently reported inhibitory actions on RyR2. Such findings prompted one possible hypothesis suggesting a primary role for RyR2 in the anti-arrhythmic actions of flecainide [6,7,16,17,18]. It was therefore of interest to compare the effects of flecainide on RyR2 channels from RyR2-P2328S mice and their WT counterparts.

## 2. Materials and Methods

### 2.1. Harvesting of Mouse Hearts 

WT and homozygous RyR2-P2328S inbred 129/Sv mice, age matched across 3 to 7 months in order to obtain a range across which would represent a similar distribution in the human population, were killed by cervical dislocation in licensed institutional premises under the UK Animals (Scientific Procedures) Act 1986. The hearts were rapidly excised and transferred to an ice cold Krebs-Henseleit buffer (in mM: NaCl 119, NaHCO_3_ 25, KCl 4, KH_2_PO_4_ 1.2, MgCl_2_ 1, CaCl_2_ 1.8, glucose 10 and Na-pyruvate 2; pH 7.4, 95% O_2_/5% CO_2_) to rinse and remove excess tissue and blood. The whole heart was then snap frozen in liquid N_2_. Hearts were couriered to Australia on dry ice and then stored at −80 °C.

### 2.2. Isolation of the RyR2 SR Vesicle Preparation 

All steps of the SR vesicle preparation were performed on ice and/or at 4 °C. Five to 7 hearts were homogenised in a cardiac homogenising buffer (CHB, containing in mM: sucrose 290, imidazole 10, and NaN_3_ 3, pH 6.9). The homogenate was then centrifuged at 12,000× *g* for 20 min, then the pellet discarded and the supernatant centrifuged at 43,000× *g* for 2 h. The pellet was resuspended in Buffer A (CHB plus 649 mM KCl) and centrifuged at 46,000× *g* for 1.5 h. The resulting pellet was resuspended in 125 µL of buffer A per gm of mouse heart plus a protease inhibitor mixture and stored in 8 μL aliquots at −80 °C. All individual protease inhibitors were obtained from Sigma Aldrich (Castle Hill NSW, Australia) and were added to the final suspension at the following final concentrations: benzamidine hydrochloride hydrate (catalogue # B6506) 1.0 mM; Pepstatin A (catalogue # P4265) 2.1 µM; Leupeptin (catalogue # L2884) 1 µM; AEBSF/Pefabloc SC (catalogue # 76307) 0.5 mM; Calpain Inhibitor I (catalogue # A6185) 3 µM; Calpain Inhibitor II (Catalogue # A6060) 3 µM.

### 2.3. Single Channel Lipid Bilayer Recordings 

Lipid bilayers were formed as previously described [5] by spreading a lipid mixture (phosphatidylethanolamine, phosphatidylserine and phosphatidylcholine in *n*-decane) across a 100 µm aperture in a partition separating the *cis* chamber and *trans* chambers. SR vesicles were added to the *cis* solution so that, following incorporation, the cytoplasmic surface of SR and RyR2 faced that solution, which was therefore equivalent to the cytoplasmic solution. SR vesicles were incorporated using a cytoplasmic (*cis*) incorporation solution containing, in mM: 230 caesium methanesulfonate (CsMS); 20 CsCl; 1 CaCl_2_ and 10 tetraethylsulfamide (TES) pH 7.4. The luminal (*trans*) incorporation solution contained, in mM: 30 CsMS; 20 CsCl; 10 TES; pH 7.4, with a physiological luminal Ca^2+^ concentration of 1 mM. Following channel incorporation, CsMS was added to the *trans* solution to equalise the concentration of the charge carrier in both solutions. Continuous current recording began at this stage and continued for the duration of the experiment. The *cis* solution was then exchanged with a recording solution, identical to the *cis* incorporation solution, except that the cytoplasmic Ca^2+^ concentration was 1 μM. The solution was exchanged using a back-to-back 10 mL syringe aspiration-perfusion system designed to effectively replace the entire *cis* bathing solution. The Ca^2+^ concentration of the perfusion solution was set by adding appropriate concentrations of BAPTA (determined using a Ca^2+^ electrode). The *cis* solution was supplemented with flecainide to final concentrations of 0.5, 1, 5, 10, 50 and 100 µM. Experiments were performed at a room temperature of 19 ± 1 °C. Note that the *trans* (luminal) Ca^2+^ concentration was maintained at its physiological level of 1 mM throughout. Note also that the initial cytoplasmic Ca^2+^ concentration of 1 mM used for vesicle incorporation was higher than the cellular range of 100 nM to 10 µM, but this was required to facilitate channel incorporation.

### 2.4. Single Channel Lipid Bilayer Electrophysiology and Analysis 

Electrodes in the solutions on either side of the bilayer were used to voltage clamp the bilayer potential to −40 or +40 mV (V*cis—*V*trans*) and to detect current flow through the channel. The bilayer potential was switched between −40 mV and +40 mV every 30 s. The open probability (*P_o_*), mean open time (*T_o_*), and mean closed time (*T_c_*) were measured over 60 to 90 s of recording in which only one channel opened in the bilayer, using the programs Channel 2 (developed by PW Gage and M Smith, JCSMR) or Channel 3 (developed by NW Laver, University of Newcastle). Threshold levels for channel opening were set to exclude baseline noise at ∼20% of the maximum single channel conductance.

### 2.5. Statistics 

The significance of differences between various parameters for all channel data were assessed with Students *t*-test. Values are expressed as means ± standard errors of the mean (SEM). Differences were considered significant with *p* < 0.05. 

## 3. Results

Flecainide was applied to RyR2 channels incorporated into lipid bilayers over a range of increasing drug concentrations from 0.5 to 100 μM (Methods). Channels were exposed to each flecainide concentration for 2 to 3 min and 60 s to 90 s of channel activity analysed at each potential. Neither ATP nor Mg^2+^ were included in the bathing solutions.

### 3.1. Overall Actions of Flecainide on WT and P2328S Channels

The control activity of channels is shown in Figure 1A. As previously reported, mutant channels were more active than WT, with both more active at −40 mV than +40 mV [5]. Unexpectedly, the activity of both the WT and P2328S channels increased relative to control with 5 µM flecainide, then returned towards control with 50 µM flecainide. There was no voltage dependence in this action of flecainide in the channel records (Figure 1A) or in channel gating parameters (Figure 1B). Therefore, values from both potentials are included in all subsequent average data. Average control *P_o_* for WT channels of 0.03 ± 0.01 was significantly less than 0.096 ± 0.024 for P2328S channels, with the P2328S channels having significantly shorter, closed durations and greater event frequency (130 ± 22.9 ms and 26.02 ± 4.97 s^−1^, respectively) compared to the WT channels (324 ± 64.0 ms and 11.00 ± 4.28 s^−1^, respectively). Furthermore, the average *P_o_* in the P2328S channels remained significantly higher than the WT with flecainide concentrations of 1.0 to 50 µM and was maintained through shorter closures and higher opening frequencies.

Curiously, the increase in *P_o_* shown by the individual channels, illustrated in Figure 1A, is not reflected in the mean parameter values in Figure 1B, although activity of many channels, particularly WT with a low control *P_o_* increased with 0.5 and 1.0 µM flecainide. In addition, the mean closed times tended to decrease and the event frequency to increase in the WT channels with 0.5 and 1.0 µM flecainide, changes that would be associated with increased activity. The decline in activity with higher flecainide concentrations, especially apparent in channels with high control *P_o_* values, was reflected in a tendency for *P_o_* to decrease, associated with significantly longer closed times, consistent with flecainide inhibition of RyR2. On the other hand, the frequency of openings in the P2328S channels tended to increase with 10 µM and 50 µM flecainide, a change not consistent with the decrease in *P_o_*.

Apparent inconsistencies between the mean values in Figure 1B and individual channel data could arise because of the wide distribution of parameter values here (Figure 1B) and reported in mouse, dog and human RyR2 [5,19,20]. With this wide range, the average values became biased towards larger values. Accordingly, a more representative indication of flecainide’s actions on individual channels was obtained by expressing values relative to the internal control for each channel before exposure to the drug [21,22] (Figure 2A).

### 3.2. Normalisation Reveals a Consistent Increase in Channel Activity with Flecainide

Consistent significant changes with flecainide emerged in the normalised data (Figure 2A). The relative *P_o_* in the WT and P2328S channels increased significantly, reaching a maximum between 1 to 5 µM and then decreasing as the drug concentrations were further increased. The relative *P_o_* then declined to control levels with 50 µM of flecainide in the WT and to significantly less than control levels in the P2328S channels with 100 µM of flecainide. The increases in *P_o_* were associated with increases in event frequency.

An apparent influence of control activity on activation by flecainide was noted in Figure 1 above. To explore this further, the maximum *P_o_* after exposure to flecainide was plotted against the control *P_o_* for each channel *(*Figure 2B). *P_o_* increased above the SEM on control activity (~1.3-fold) in 78% of WT channels with a low control *P_o_* but did not increase in any channels with a *P_o_* greater than 0.08 (Figure 2B(I)). Thus *P_o_* = 0.08 formed a ceiling above which there was no increase in WT activity. In contrast, there was no obvious ceiling for increased activity in the P2328S channels (Figure 2B(II)). To explore gating changes associated with increased or decreased activity, data was therefore sorted into high and low activity groups using a threshold control *P_o_* of 0.08. 

### 3.3. The Control Activity of Individual Channels Determines the Action of Flecainide

Examples of channels with a high control *P_o_* are shown in Figure 3A(I,II). The average relative *P_o_* in high activity WT channels did not change with the introduction of 0.5 and 1.0 μM of flecainide but fell significantly with flecainide concentrations of 5 to 50 µM (Figure 3B(I)). The average relative *P_o_* in the P2328S declined significantly with concentrations of 50 and 100 µM of flecainide (Figure 2B(II)). Notably, the relative *P_o_* in the WT channels with 50 μM of flecainide was significantly less than the *P_o_* of the P2328S channels with 50 or 100 μM of flecainide. The decline in *P_o_* was associated with a significant abbreviation of open times. WT channel closures became longer and open frequency declined consistent with a reduced *P_o_* and reduced open duration. In contrast, the P2328S channels exhibited a small but significant decrease in closed durations and increased open frequency with higher flecainide concentrations, both opposing the reduced open time and contributing to significantly smaller declines in *P_o_*. Average parameter values (Table 1) generally reflect the changes in normalised values.

In marked contrast to the higher activity channels, an increase in *P_o_* dominated in low activity channels (Figure 4A). *P_o_* peaked with 5 and 10 µM flecainide, then declined as concentrations increased and became significantly less than the control values in the P2328S, with concentrations reaching 100 μM flecainide. Event frequency changed with *P_o_* particularly in the mutant channels, but there were no consistent changes in open or closed times in either channel type. Again, changes in the average parameter values generally reflect the changes in normalised values (Table 2).

The results in Figure 2 and Figure 3 indicate that an increase in activity was associated with an increase in opening frequency and reduced closed time, while the decline in activity was associated with a reduced open time and event frequency.

## 4. Discussion

We report for the first time significant increases in the activity of isolated RyR2 channels from WT and P2328S mouse hearts following an application of low cytoplasmic concentrations (0.5 to 10 µM) of flecainide. However, with further increased concentrations, activity declined to significantly below control levels at 50 and 100 µM flecainide. This result was consistent with flecainide binding to an activation site, dominating the response at low flecainide concentrations, while binding to an inhibitory site with increased concentrations, which reduced or even reversed this activation effect. The activation was shifted to higher flecainide concentrations, reaching a peak between 5 and 10 µM in P2328S, compared to 1 to 5 µM in WT channels. Since *P_o_* reflects the summation of both activation and inhibition, these differences could be explained by either stronger activation or weaker inhibition in P2328S compared to WT. It was surprising that the P2328S channels were not more inhibited by flecainide than the WT, given that those channels were generally more active than WT channels. However, this was consistent with activation dominating the action of flecainide on mutant channels, possibly caused by changes in the activating binding site as a result of the P2328S mutation. It should be reiterated that the aim of the present study was to decipher the basic action of flecainide on RyR2 channel gating. It would not be surprising if the flecainide’s action differed between isolated RyR2 channels and RyR2 channels in myocytes, where the channels are subject to a multitude of other modulatory influences. This is illustrated by the apparent dominance of activation in the P2328S channels in contrast to flecainide inhibition of RyR2-mediated Ca^2+^ release in the P2328S mouse [7]. 

These observations on isolated channels were obtained under conditions in which currents through the individual channels were recorded for up to 90 min, with channel parameters generally measured over 60 to 90 s at each flecainide concentration. The two to ten-fold changes in channel activity with flecainide were maintained for many minutes. RyR2 activity does not spontaneously change during bilayer experiments in the absence of specific interventions, although *P_o_* can fluctuate by 10 to 20% over several minutes [23,24,25]. That RyR2 activity does not spontaneously increase or decrease during experimentation is evidenced by changes in *P_o_* with specific interventions being fully reversed when the intervention is removed [26,27].

RyR2 activation by flecainide is a novel and unexpected, hitherto unreported, finding. It should be noted however that the present experimental conditions differed from those of previous studies. First, a number of earlier studies were made under conditions producing high *P_o_* values (0.89–0.99) close to the 1.00 ceiling in the presence of 1 to 2 mM of cytoplasmic ATP [16,21,28]. These almost maximal *P_o_* values could mask flecainide-mediated activation as *P_o_* is unable to increase further even if there are further activating conformational changes in the protein. However, a simple ceiling effect might not be the reason, because we did not see activation in our high activity RyR2 channels where an average *P_o_* between 0.12 and 0.25 would have permitted a >two-fold increase in *P_o_* in the presence of a flecainide mediated activation mechanism. In addition, some previous experiments were performed in the presence of Mg^2+^ to lower *P_o_* to more physiological levels while in the presence of ATP [21]. Inclusion of either or both ATP and Mg^2+^ could potentially also reduce the activating effect of flecainide. In this context, Bannister et al. [17], using only 2,3-butanedione monoxime (BDM-) 4100 (C_14_H_13_N_3_OS)) to increase *P_o_* to 0.98 before flecainide addition, show weak trends towards decreased closed times with flecainide, consistent with changes also seen in the high activity of the P2328S channels, possibly due to the presence of an activating influence. An additional factor that could contribute to activation being overlooked may be flecainide concentration. We show the strongest effects of activation with 0.5 to 1.0 µM flecainide, concentrations not generally explored in most previous studies which mainly examine the effects of flecainide with concentrations ≥5 µM, and showed reduced activities at ≥10 µM [16,21,28], similar to trends seen in the present observations.

In contrast to those previous studies [16,17,21], and as a result of the minimal experimental conditions in the present experiments, the RyR2 channels were studied in an absence of other cytosolic regulatory binding factors such as ATP and Mg^2+^ [16,21]. This minimised the number of potential interactions that might modify the structure of the RyR2 and mask the effects of flecainide alone on channel activity, in other words that might reveal the direct effects of flecainide on RyR2 gating. However, as with previous studies, the channels in the present experiments maintained the vesicular RyR2 associations with binding proteins, such as triadin, junctin, calsequestrin 2, junctaphilin and FKBP12 [5,29,30]. Importantly, the control *P_o_* values in the present study (0.001 to 0.4) covered much of the physiological range. Systolic *P_o_* is ~0.4 [31], as in the high activity channels. Diastolic *P_o_* is predicted to be 0.00001 to 0.00004 [32], too low for accurate single channel measurements. Our low activity control *P_o_* (0.001 to 0.08) would however exist in late diastole particularly in CPVT and in early systole.

Together the present and previous findings suggest possible dual activating and inactivating effects of flecainide on RyR2. At the cellular level, this could be reflected in differing reports of its in vivo actions in both suppressing Ca^2+^ waves in intact and permeabilized myocytes, while increasing Ca^2+^ spark frequency [28]. In contrast, in intact hearts from a homozygotic CPVT mouse model exhibiting compromised Nav1.5 function, flecainide, despite being a Nav1.5 blocker, mediated anti-arrhythmic effects likely via the inhibition of RyR2-mediated Ca^2+^ release [6,7,16,33]. This indirect, RyR2-mediated effect may be the basis for the recent use of flecainide monotherapy in human CPVT, although here individuals are usually heterozygotic and thus flecainide could be acting on both WT and mutant RyR2 [34,35,36]. Flecainide’s direct pro-arrhythmic actions may continue to reinforce the contra-indication of its use with structural heart disease [16,37], which has been attributed to its direct Nav1.5 inhibitory effect and would be exacerbated by the RyR2 activation reported here. The findings together are consistent with RyR2 being a large molecule influenced by multiple activation and inhibition sites potentially targeted by flecainide and variously located in RyR2’s peripheral and central cytoplasmic and luminal domains all converging on gating mechanisms near the channel pore. The voltage independence of the flecainide-mediated activation reported here would suggest a binding site remote from the membrane in contrast to voltage-dependent inhibition, which suggests a site influenced by the membrane field [21]. Residue P2328 is situated on the HD1 Helical Domain, which has long range interactions with gating regions near the RyR2 channel pore [38,39,40], therefore compatible with a mutation that modifies the activating response to flecainide.

## 5. Conclusions

There is an activation site for flecainide on the RyR2 channel that may not be accessible under most cellular conditions but may become exposed under particular experimental or clinical situations. Flecainide activation at this site may contribute to reports of occasional paradoxical proarrhythmic actions of flecainide. Nevertheless, the inhibitory action likely dominates in high open probability RyR2 channels, contributing to its clinical anti-arrhythmic efficacy in CPVT. Overall, the results underscore the importance of understanding the underlying cause of an arrhythmia to ensure an appropriate therapeutic strategy.

## Figures and Tables

**Figure 1 cells-10-02101-f001:**
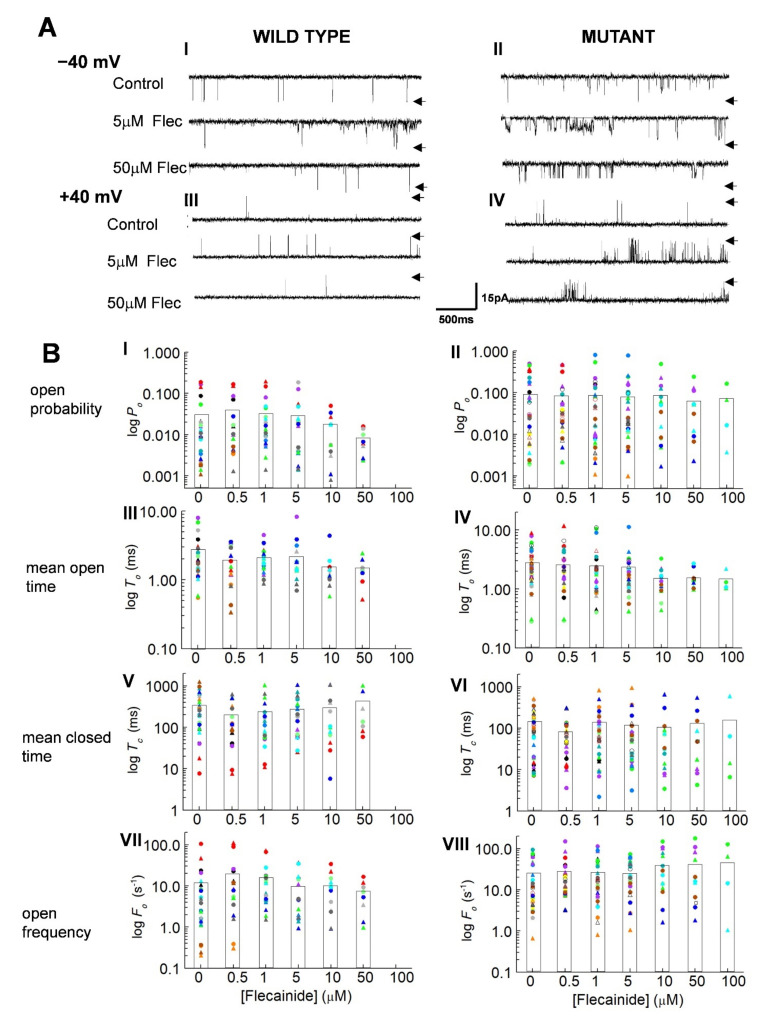
Actions of flecainide on individual wild type (WT) and P2328S RyR2 channels. (**A**) Channel currents before addition of flecainide (upper) and with 5 µM flecainide (middle) and 50 µM flecainide (lower) at −40 mV (**A**) (I,II) and +40 mV (**A**) (III,IV). Arrows indicate maximum open current. (**B**) Individual data for WT (*n* = 13) and P2328S (*n* = 17) channels. Relative open probability (*P_o_*) (**B**) (I,II), mean open time (*T_o_*) (**B**) (III,IV), mean closed time (*T_c_*) (**B**) (V,VI) and open frequency (*F_o_*) (**B**) (VII,VIII) at +40 mV (circles) and −40 mV (triangles). Data for each channel is represented by symbols of the same colour. The bar graphs show the mean values for each of the parameters at each flecainide concentration.

**Figure 2 cells-10-02101-f002:**
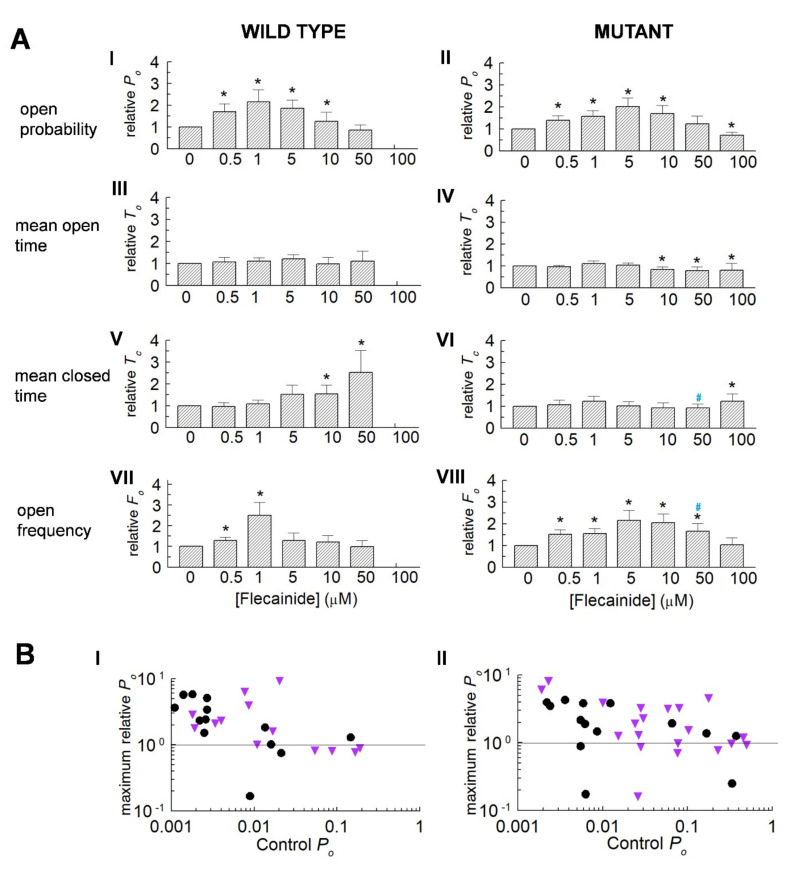
Actions of flecainide on relative parameter values for wild type (WT) and P2328S RyR2 channels. (**A**) Average normalised data (mean ± SEM) for WT (*n* = 13) and P2328S (*n* = 17) channels. Relative open probability (*P_o_*) (**A**) (I,II), mean open time (*T_o_*) (**A**) (III,IV), mean closed time (*T_c_*) (**A**) (V,VI) and open frequency (*F_o_*) (**A**) (VII,VIII). Asterisks (*) indicate significant (*p* < 0.05) changes from control. Blue ^#^ indicates significant (*p* < 0.05) differences between P2328S and WT. (**B**) Individual maximum relative *P_o_* values with flecainide, plotted against control *P_o_* for WT (**B**) (I) and P2328S (**B**) (II). Black circles, +40 mV; purple triangles, −40 mV. Maximum relative *P_o_* is the highest value recorded with flecainide. Values below 1 indicate no increase in activity above control, but do not reflect the full decline in activity.

**Figure 3 cells-10-02101-f003:**
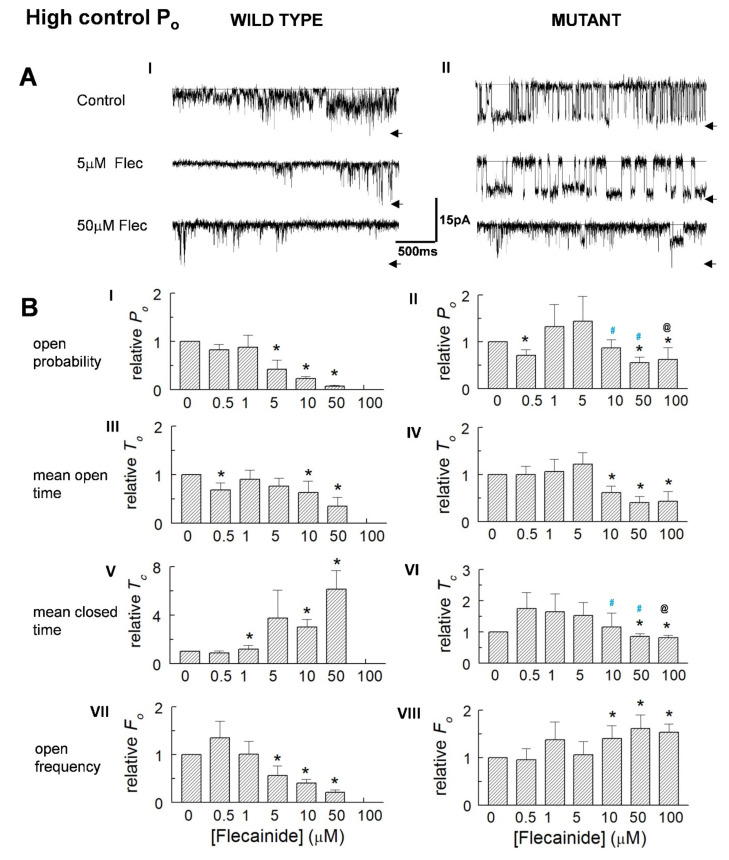
Reduced activity of high control *P_o_* WT and P2328S channels. (**A**) Currents at −40 mV from channels with control *P_o_* = 0.188 (wild type (WT), (**A**) (I) or 0.494 (P2328S, (**A**) (II)), before flecainide (upper) and with flecainide at 5 µM (middle) and 50 µM (lower). Arrows indicate maximum open current. (**B**) Relative open probability (*P_o_*) (**B**) (I,II), mean open time (*T_o_*) (**B**) (III,IV), mean closed time (*T_c_*) (**B**) (V,VI) and open frequency (*F_o_*) (**B**) (VII,VIII). Data is mean ± SEM. Asterisk (*), significant (*p* < 0.05) change from control. Blue ^#^, significant (*p* < 0.05) difference between P2328S and WT, ^@^ significant (*p* < 0.05) differences between 0.5 or 1 µM and 100 µM flecainide. WT (*n* = 3) and P2328S (*n* = 6).

**Figure 4 cells-10-02101-f004:**
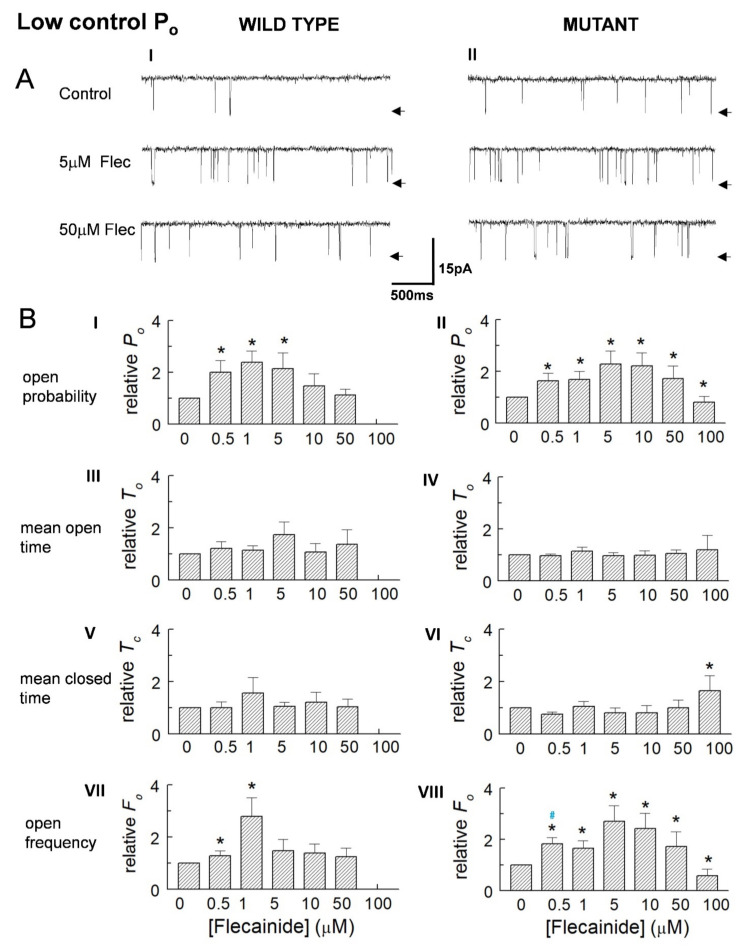
Increased activity of low control open probability (*P_o_*) wild type (WT) and P2328S channels. (**A**) Average relative *P_o_* (**A**) (I,II), mean open time (*T_o_*) (**A**) (III,IV), mean closed time (*T_c_*) (**A**) (V,VI) and open frequency (*F_o_*) (**A**) (VII,VIII) for WT (*n* = 10) and P2328S (*n* = 11). (**B**) Average parameter values for WT (**B**) (I) and P2328S (**B**) (II). WT channels were not exposed to 100 µM flecainide. In (**B**) data is mean ± SEM. (*), significant (*p* < 0.05) change from control. Blue ^#^, significant (*p* < 0.05) difference between WT and P2328S.

**Table 1 cells-10-02101-t001:** Gating parameters for high or activity wild type (WT) and P2328S RyR2 channels before and during flecainide exposure. Parameter values for WT and P2328S channels before adding flecainide ([Flec] = 0 μM) and with 0.5 μM to 50 μM (WT) or 100 μM (P2328S) flecainide. 100 μM flecainide not applied to WT channels (nd, not done). Mean ± SEM is presented. *P_o_*, open probability; *T_o_*, mean open time; *T_c_*, mean closed time; *F_o_*, frequency of opening. Asterisk (*) denotes significant (*p* < 0.05) difference from control. Blue **^#^** denotes P2328S showing significant (*p* < 0.05) differences from WT.

[Flec]	0 µM	0.5 µM	1.0 µM	5 µM	10 µM	50 µM	100 µM
WILD TYPE							
*P_o_*	0.147 ± 0.022	0.120 ± 0.024	0.142 ± 0.034	0.067 ± 0.032 *	0.039 ± 0.004 *	0.011 ± 0.005 *	nd
*T_o_* (ms)	4.18 ± 0.34	2.43 ± 0.05	3.14 ± 0.69	3.71 ± 2.28	1.37 ± 0.17	0.73 ± 0.22 *	nd
*T_c_* (ms)	26.65 ± 8.23	23.26 ± 8.62	25.17 ± 13.28	48.75 ± 11.93	35.39 ± 7.75	70.5 ± 11.6 *	nd
*F_o_* (s^−1^)	49.32 ± 19.67	62.34 ± 22.43	52.41 ± 17.40	22.57 ± 7.20	28.35 ± 5.70	14.40 ± 2.31	nd
P2328S							
*P_o_*	0.241 ± 0.044 ^#^	0.204 ± 0.065	0.227 ± 0.079 ^#^	0.228 ± 0.083	0.201 ± 0.087 ^#^	0.133 ± 0.040 *^,#^	0.116 ± 0.049
*T_o_* (ms)	4.08 ± 0.72	4.01 ± 1.22	3.44 ± 1.09	3.98 ± 1.07	1.97 ± 0.40 *	1.23 ± 0.14 *	1.17 ± 0.13 *
*T_c_* (ms)	15.07 ± 2.78 ^#^	19.39 ± 4.70	24.55 ± 12.83	20.92 ± 7.86 ^#^	11.43 ± 3.62 ^#^	10.16 ± 2.83 ^#^	10.50 ± 3.91
*F_o_* (s^−1^)	60.58 ± 6.25	61.72 ± 14.74	66.00 ± 10.24	49.98 ± 6.78 ^#^	92.89 ± 18.95 *^,#^	105.80 ± 26.53 *^,#^	95.76 ± 31.07 *

**Table 2 cells-10-02101-t002:** Gating parameters for high or activity wild type (WT) and P2328S RyR2 channels before and during flecainide exposure. Parameter values for wild type (WT) and P2328S channels before adding flecainide ([Flec] = 0 μM) and with 0.5 μM to 50 μM (WT) or 100 μM (P2328S) flecainide. 100 μM flecainide not applied to WT channels (nd, not done). Mean ± SEM is presented. *P_o_*, open probability; *T_o_*, mean open time; *T_c_*, mean closed time; *F_o_*, frequency of opening. Asterisk (*) denotes significant (*p* < 0.05) difference from control. Blue ^#^ denotes P2328S showing significant (*p* < 0.05) differences from WT.

[Flec]	0 µM	0.5 µM	1.0 µM	5 µM	10 µM	50 µM	100 µM
WILD TYPE							
*P_o_*	0.009 ± 0.003	0.009 ± 0.002	0.015 ± 0.012	0.023 ± 0.012	0.011 ± 0.003	0.007 ± 0.002	nd
*T_o_* (ms)	2.01 ± 0.31	1.63 ± 0.31	1.86 ± 0.18	4.43 ± 2.66	1.50 ± 0.34	1.67 ± 0.19	nd
*T_c_* (ms)	387.4 ± 69.7	254.7 ± 57.6	249.1 ± 61.0	335.6 ± 71.4	355.3 ± 116.2	413.8 ± 154.7	nd
*F_o_* (s^−1^)	4.03 ± 0.81	5.84 ± 1.24	8.82 ± 1.79 *	7.73 ± 2.52	6.54 ± 1.61	4.61 ± 1.34	nd
P2328S							
*P_o_*	0.017 ± 0.005 ^#^	0.028 ± 0.010 ^#^	0.026 ± 0.013	0.025 ± 0.010	0.028 ± 0.018	0.019 ± 0.011	0.010 ± 0.011
*T_o_* (ms)	2.08 ± 0.28	2.09 ± 0.41	2.03 ± 0.52	1.46 ± 0.05	1.15 ± 0.14 *	1.67 ± 0.29	1.64 ± 0.57
*T_c_* (ms)	192.4 ± 27.16 ^#^	113.8 ± 20.71 *^,#^	176.3 ± 43.20	152.6 ± 52.76 ^#^	154.4 ± 69.40	207.8 ± 114.9	331.2 ± 267.5
*F_o_* (s^−1^)	7.172 ± 0.935 ^#^	12.79 ± 2.117 *^,#^	9.96 ± 1.570	16.26 ± 3.074 *^,#^	20.54 ± 7.924 *^,#^	10.92 ± 4.150 *^,#^	7.715 ± 6.665

## Data Availability

Data is contained within the article.

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
