# Peer review of "Flecainide Paradoxically Activates Cardiac Ryanodine Receptor Channels under Low Activity Conditions: A Potential Pro-Arrhythmic Action"

_cells, 2021, doi:10.3390/cells10082101_

Round 1
Reviewer 1 Report
Comments to author
In this paper, Salvage et al. examined effect of flecainide on the cardiac ryanodine receptor (RyR2) channel using single channel recordings in lipid bilayer membranes with cardiac microsomes prepared from WT and RyR2-P2328S mice. They found that flecainide exhibited dual effects on the RyR2 channel: activation under low activity conditions and inhibition under high activity conditions. They propose that flecainide binds separate activation and inhibition sites on RyR2.
The results in this paper are novel and interesting. However, there are several concerns about presentation of the items.
Major revisions
The authors presented Table 1 as a summary of the experiments. However, as they stated in the paper, the RyR2 channel exhibits large heterogeneity. Therefore, mean values in Po are less meaningful. Instead, the reviewer strongly recommend that authors show bar graphs for Po with mean and individual values (not SD or SE) for all channels under respective conditions. This will greatly help understand the range of heterogeneity of the Po.
Figure 1C shows summary of Po values and the maximum relative values with flecainide. This is useful for activation of the channel by flecainide, but not applicable for inhibition by the drug. Please also show "Po values and the MINIMUM relative values with flecainide". Also, since no voltage dependence is already shown in Fig. 1A, I recommend that the averaged values are used for the plot in Fig. 1C and the new plot. These items will demonstrate that low control Po channels are activated by flecainide, whereas high control Po channels are not activated but inhibited by the drug.
Since Fig. 2C and 3B show absolute values, but essentially the same with Fig. 2B and 3A, respectively. Please delete Fig. 2C and 3B. If you really want to show them, please remove them from the figures and add in new Table (Table should not be included in the figure).
Minor revisions
Please provide number of channels in Fig. 2.
Please provide typical traces for low control Po channels in Fig. 3.
A paragraph in Discussion (starting with "In contrast to those...) is duplicated. Please delete it.
Author Response
We thank the reviewers for their positive comments and suggestions, most of which we have included and which have improved the manuscript. Our detailed responses to each of the reviewers are detailed below.
Reviewer 1.
Point 1. The authors presented Table 1 as a summary of the experiments. However, as they stated in the paper, the RyR2 channel exhibits large heterogeneity. Therefore, mean values in Po are less meaningful. Instead, the reviewer strongly recommend that authors show bar graphs for Po with mean and individual values (not SD or SE) for all channels under respective conditions. This will greatly help understand the range of heterogeneity of the Po.
Response. We have replotted the data as suggested in the new Figure 1B, which now also contains the representative channel recordings (Figure 1A). The graphs in Figure 1B show the data from each channel presented in a particular colour to allow individual channels to be followed through the concentration range. The mean for each concentration is indicated by bars in each graph.
Point 2. Figure 1C shows summary of Po values and the maximum relative values with flecainide. This is useful for activation of the channel by flecainide, but not applicable for inhibition by the drug. Please also show "Po values and the MINIMUM relative values with flecainide". Also, since no voltage dependence is already shown in Fig. 1A, I recommend that the averaged values are used for the plot in Fig. 1C and the new plot. These items will demonstrate that low control Po channels are activated by flecainide, whereas high control Po channels are not activated but inhibited by the drug
Response. BTW Figure 1C in the original manuscript is now Figure 2B in the revised manuscript. The components of Point 2 have been addressed indirectly, while the graph has not been changed for several reasons outlined below.
First, the minimum Po values for each channel is now shown in the new Figure 1B and are more appropriate in that graph of Po vs flecaininde concentration than in the graph of max Po (at any concentration) vs starting control Po now the new Figure 2B. The aim of the graph in new Figure 2B (original Figure 1C) is to address whether or not channels are activated by flecainide at any concentration. Activation was flagged in the description of Figure 1A as follows: “Unexpectedly, the activity of both WT and P2328S channels increased relative to control with 5 µM flecainide, then returned towards control with 50 µM flecainide” and of Figure 1B as follows: “Curiously, the increase in Po shown by individual channels, illustrated in Figure 1A, is not reflected in the mean parameter values in Figure 1B, although activity of many channels, particularly WT with a low control Po increased with 0.5 and 1.0 µM flecainide”. It is now stated specifically in the revised text in the second paragraph under point 3.2, as follows: “An apparent influence of control activity on activation by flecainide was noted in Figure 1 above. To explore this unexpected activation further, the maximum Po after exposure to flecainide was plotted against the control Po for each channel (Figure 2B).”
Second, since as shown in Figure 1A and 1B activity in many channels increased at low concentrations, but declined at higher concentrations, it is not clear how the minimum Po could be represented simply in the graph of maximum Po for each channel as a function of the starting control Po, without changing the context and significance of figure 2B. We hope that the data in revised Figure 1B will address this point.
Third, the fact that the activation by flecainide is independent of voltage is a major factor supporting the suggestion that there are separate activation and inhibition sites, because inhibition reported previously is strongly voltage-dependent. Voltage-dependent effects are likely due to flecainide binding to sites influenced by (and likely close to) the membrane, whereas voltage-independent effects are likely mediated by sites remote from the membrane (mentioned in the Discussion). Therefore it is important to show that voltage-independence is apparent in all channels, and independent of how the data is analysed. Justifying this observation by referring to observations in only 2 individual channels in the Figure 1A (revised and original figures) does not illustrate the generality of the phenomena. Consequently separate values are shown for +40mV and -40mV in both Figure 1B and with the different analysis in Figure 2B. As suggested by the reviewer, the average of data at +40mV and -40mV are included in all other cases.
Point 3. Since Fig. 2C and 3B show absolute values, but essentially the same with Fig. 2B and 3A, respectively. Please delete Fig. 2C and 3B. If you really want to show them, please remove them from the figures and add in new Table (Table should not be included in the figure).
Response. The original figures 2C and 3B have been deleted as suggested and the data is now shown in Table 1 and Table 2 respectively. It is useful to show this data as (1) it shows that the trends in the measured parameters are similar to significant changes in the normalised data and (2) reinforces the use of normalization to reveal significant changes in channel activity.
Minor revisions.
- Please provide number of channels in Fig. 2 (new Figure 3).
- Please provide typical traces for low control Po channels in Fig. 3 (new Figure 3). Done
- A paragraph in Discussion (starting with "In contrast to those...) is duplicated. Please delete it. Done
General
Both reviewers comment of different writing styles as a result of the contribution of the various authors. Numerous small changes have been made to address this comment.
Reviewer 2 Report
Salvage et al. reported that Flec preferably activates RyR with low intrinsic activity, while inhibits in channels with high intrinsic activity. Findings from this study have direct clinical relevance in the treatment of arrhythmogenic cardiomyopathies. Overall the authors are to be commended for performing an interesting study. Although the findings from this study add granular details into the understanding of anti-arrhythmic drugs, some findings are very difficult to interpret due to numerical discrepancies. The translatability of the study is also very limited, while some physiological readouts to support the ventricular pro-arrhythmic action are necessary.
Major comments
1- If “Figure 1A was representative of the average values shown in Table 1 (Lines 131-132)”, how “The increase in Po shown by individual channels, illustrated in Figure 1A, is not reflected in the average parameter values in Table 1 (Lines 148-149)”? Although the authors tried to articulate an explanation, this still is very confusing. Authors claim the wide range variability as a possible reason; however, the mean, SEM, and range in WT 0 uM vs WT 5 uM (Table 1) are identical but WT 5 uM shows up as significant in Fig 1B. Considering the title of this study, this concern raises questions about whether Flec activates RyR or if this observation is an artificial event based on mathematical calculations. A statistician consultation is recommended.
2- Kryshtal et 2021 (cited by the authors), recently demonstrated a modest decrease in the Po of RyR and no pro-arrhythmic Ca waves using 5 uM, while higher concentrations markedly inhibited the Po and suppressed Ca waves. Thus, taking into consideration previous proof-of-concept studies demonstrating the anti-arrhythmic effects of flec and the clinical efficacy in patients with inherited cardiac arrhythmias (including CPVT), the proposed “Potential Pro-arrhythmic Action” remains largely conjectural.
3- Although I agree that each RyR behaves according to its individual gating properties and appreciate the efforts to create a cut-off to distinguish Flec effects on channels with low and high Po, it is not clear would impact the overall changes in isolated cardiomyocytes. Addressing this concern is important to reveal a physiological relevance of low intrinsic RyR activity on the “Potential Pro-arrhythmic Action” of Flec.
4- Also, considering that the majority of mutated RyR are more active and the proposed concept that flec inhibits the subset of high Po channels, a more pronounced inhibitory effect was expected in mutated RyR. A discussion is recommended.
Minor comments
- 2 different Ref styles are used.
- A creative graphical abstract “digesting” the main findings would facilitate the understanding for non-EP scientists.
Author Response
We thank the reviewers for their positive comments and suggestions, most of which we have included and which have improved the manuscript. Our detailed responses to each of the reviewers are detailed below.
Reviewer 2.
Point 1. - If “Figure 1A was representative of the average values shown in Table 1 (Lines 131-132)”, how “The increase in Po shown by individual channels, illustrated in Figure 1A, is not reflected in the average parameter values in Table 1 (Lines 148-149)”? Although the authors tried to articulate an explanation, this still is very confusing. Authors claim the wide range variability as a possible reason; however, the mean, SEM, and range in WT 0 uM vs WT 5 uM (Table 1) are identical but WT 5 uM shows up as significant in Fig 1B. Considering the title of this study, this concern raises questions about whether Flec activates RyR or if this observation is an artificial event based on mathematical calculations. A statistician consultation is recommended.
Response part 1. The point of normalizing data as in the original Figure 1B was to remove the variability introduced by averaging a wide range of original values (original Table 1). In other words we wanted to know whether on average the activity of channels went up or down as a function of flecainide concentration. Any effect on low activity channels was lost in the average of the actual parameter values due to the bias towards higher values. With normalization of parameter values for each individual channel with flecainide to the internal control for that channel before application of flecainide, significant changes that reflected the changes apparent in the original channel records (Figure 1A, 3A and 4A) are revealed and are reinforced with further separation into high activity and low activity groups which revealed that high activity channels did not show the same response to flecainide as the low activity channels. We hope that this has been clarified by the graphs showing changes in the activity of individual channel activity in the new Figure 1B. As stated in the manuscript, normalisation of single RyR channel data to internal control values is a commonly used strategy including publications from our laboratory [21] and others [22].
The reviewers observation that, the mean, SEM, and range in WT 0 uM vs WT 5 uM (Table 1) are identical but WT 5 uM shows up as significant in Fig 1B, is precisely point of normalising the data.
Response part 2. The fact that individual low activity channels show an increase in activity particularly at low flecainide concentrations (Figures 1A and 1B and Figure 4A) cannot be attributed to an artificial event based on mathematical calculations. These are original recordings of channel activity with no mathematical or statistical manipulation. The fact that the increase in activity was not seen in the average data in the original Table 1 and in the mean values (bar graphs) in the new Figure 1B is a statistical artifact of averaging data from two effectively independent populations of channels. We have attempted to separate the two populations by separating the data into high activity and low activity groups. This was clarified in the last paragraph under 3.2.
Point 2. Kryshtal et 2021 (cited by the authors), recently demonstrated a modest decrease in the Po of RyR and no pro-arrhythmic Ca waves using 5 uM, while higher concentrations markedly inhibited the Po and suppressed Ca waves. Thus, taking into consideration previous proof-of-concept studies demonstrating the anti-arrhythmic effects of flec and the clinical efficacy in patients with inherited cardiac arrhythmias (including CPVT), the proposed “Potential Pro-arrhythmic Action” remains largely conjectural.
Response part 1. We agree in part with the reviewer and we had addressed this in the conclusions of the original manuscript in which we state that: “There is an activation site for flecainide on the RyR2 channel,that may not be accessible under most cellular conditions but may become exposed under particular experimental or clinical situations. Flecainide activation at this site may contribute to reports of occasional paradoxical pro-arrhythmic actions of flecainide.”
As an aside, not all inhibitory actions of flecainide are consistent with its anti-arrhythmic action as inhibition is reported to be strongest when the Ca2+ flux is from the cytoplasm into the SR (Bannister et al., 2015 and 2016; Salvage et al., 2018).
Response part 2. We do not agree with the second sentence in this point. Firstly as we stated in the introduction and explored in the Discussion, occasional pro arrhythmic effects of flecainide have been reported. These effects cannot be explained by a strong inhibitory effect of flecainide on RyR2, but an activating effect as seen in our data could provide a possible explanation, in addition to the classic sodium channel inhibition. The original records and average normalised data presented in the results show clearly that under our minimal conditions flecainide can activate both WT and mutant RyR2 channels. We admit we were surprised by this finding, and we do not claim that flecainide is normally pro-arrhythmic or that it activates RyR2 under cellular conditions or in the whole heart. Nevertheless, this novel finding warrants reporting given it could under certain circumstances influence the response seen on flecainide treatment and thereby contribute to the contrasting anti and pro-arrhythmic effects seen clinically. This was addressed in the Discussion and Conclusions. The point is further addressed in the first paragraph of the Discussion as follows: It would not be surprising if flecainide’s action differed between isolated RyR2 channels and RyR2 channels in myocytes, where the channels are subject to a multitude of other modulatory influences. This is illustrated by apparent dominance of activation in P2328S channels in contrast to flecainide inhibition RyR2-mediated Ca2+ release in the P2328S mouse [7].
To re-iterate, given that RyR2 activity in the cell is an integration of many activating and inhibitory influences, which can be modified by mutations or drugs, we feel that it is important to indicate that there is an action of flecainide that could become pro-arrhythmic under appropriate conditions (that might supress its inhibitory actions) become pro-arrhythmic. Such conditions could arise with novel drug testing or with effects of pro-arrhythmic mutations that might be described in future studies.
Point 3- Although I agree that each RyR behaves according to its individual gating properties and appreciate the efforts to create a cut-off to distinguish Flec effects on channels with low and high Po, it is not clear would impact the overall changes in isolated cardiomyocytes. Addressing this concern is important to reveal a physiological relevance of low intrinsic RyR activity on the “Potential Pro-arrhythmic Action” of Flec.
Response. Creating a cut-off to distinguish flecainide effects on channels with low and high Po was done in order to obtain insight into the effects on channel gating of flecainide binding the activation site and the inhibition site. This is a strong point of interest to those focussing on the molecular/biophysical changes in the RyR2 protein produced by flecainide. The results suggest that the increase in activity depends on an increase in the frequency of open events in WT and P2328S channels (Figure 4B), while the decline in activity at higher concentrations in WT channels depends primarily on a decline in open times, leading to longer periods of channel closure and reduced event frequency (Figure 3B). These different gating changes as a result of using thea cut-off Po to partially separate the different actions and thus provides strong evidence to support the hypothesis that there are separate flecainide binding sites facilitating activation and inhibition.
Point 4- Also, considering that the majority of mutated RyR are more active and the proposed concept that flecainide inhibits the subset of high Po channels, a more pronounced inhibitory effect was expected in mutated RyR. A discussion is recommended.
Response. This is now addressed at the end of the first paragraph of the Discussion as follows: “It was surprising that the P2328S channels were not more strongly inhibited by flecainide than WT, given that the mutant channels were generally more active than WT channels. However this was consistent with the dominating influence of activation on the overall effect of flecainide possibly caused by changes in the activating binding site as a result of the P2328S mutation”. The clinical significance of the inhibitory action was (and is) also discussed later in the second last paragraph of the Discussion and has been added to the Conclusions as follows “Nevertheless, the inhibitory action likely dominates in high open probability RyR2 channels, contributing to its clinical anti-arrhythmic efficacy in CPVT. This underscores the importance of understanding the underlying cause of an arrhythmia to ensure an appropriate therapeutic strategy”.
Minor points
Minor comments
Point 1 - 2 different Ref styles are used. This has been corrected
Point 2 - A creative graphical abstract “digesting” the main findings would facilitate the understanding for non-EP scientists. A graphical abstract is included.
General
Both reviewers comment of different writing styles as a result of the contribution of the various authors. Numerous small changes have been made to address this comment.
Round 2
Reviewer 1 Report
The manuscript has now been satisfactory revised.
Reviewer 2 Report
The authors satisfactorily addressed my concerns. No further comments.